# Unraveling the Diverse Profile of N-Acyl Homoserine Lactone Signals and Their Role in the Regulation of Biofilm Formation in *Porphyra haitanensis*-Associated *Pseudoalteromonas galatheae*

**DOI:** 10.3390/microorganisms11092228

**Published:** 2023-09-04

**Authors:** Muhammad Aslam, Pengbing Pei, Peilin Ye, Tangcheng Li, Honghao Liang, Zezhi Zhang, Xiao Ke, Weizhou Chen, Hong Du

**Affiliations:** 1Guangdong Provincial Key Laboratory of Marine Disaster Prediction and Prevention, College of Science, Shantou University, Shantou 515063, China; drmaslam@hotmail.com (M.A.); peipengbing1990@126.com (P.P.); 19plye@stu.edu.cn (P.Y.); tchli@stu.edu.cn (T.L.); 16hhliang@stu.edu.cn (H.L.); 20zzzhang1@stu.edu.cn (Z.Z.); 20xke@stu.edu.cn (X.K.); wzchen@stu.edu.cn (W.C.); 2Faculty of Marine Sciences, Lasbela University of Agriculture, Water and Marine Sciences, Uthal 90150, Pakistan; 3STU-UNIVPM Joint Algal Research Center, College of Science, Shantou University, Shantou 515063, China

**Keywords:** acyl homoserine lactones, biofilms, macroalgae, *Pseudoalteromonas galatheae*, *Porphyra haitanensis*, quorum sensing

## Abstract

N-acyl homoserine lactones (AHLs) are small, diffusible chemical signal molecules that serve as social interaction tools for bacteria, enabling them to synchronize their collective actions in a density-dependent manner through quorum sensing (QS). The QS activity from epiphytic bacteria of the red macroalgae *Porphyra haitanensis*, along with its involvement in biofilm formation and regulation, remains unexplored in prior scientific inquiries. Therefore, this study explores the AHL signal molecules produced by epiphytic bacteria. The bacterium isolated from the surface of *P. haitanensis* was identified as *Pseudoalteromonas galatheae* by 16s rRNA gene sequencing and screened for AHLs using two AHL reporter strains, *Agrobacterium tumefaciens* A136 and *Chromobacterium violaceum* CV026. The crystal violet assay was used for the biofilm-forming phenotype. The inferences revealed that *P. galatheae* produces four different types of AHL molecules, i.e., C4-HSL, C8-HSL, C18-HSL, and 3-oxo-C16-HSL, and it was observed that its biofilm formation phenotype is regulated by QS molecules. This is the first study providing insights into the QS activity, diverse AHL profile, and regulatory mechanisms that govern the biofilm formation phenotype of *P. galatheae*. These findings offer valuable insights for future investigations exploring the role of AHL producing epiphytes and biofilms in the life cycle of *P. haitanensis*.

## 1. Introduction

The macroalgae are not only important photosynthetic ecosystem engineers in the marine environment, but they also contribute to the overall productivity of the oceans by fixing carbon dioxide [1,2]. Moreover, they serve as biological indicators of ecosystem health in monitoring programs worldwide [3]. The surfaces of macroalgae are diverse, complex, and ever-changing, making them ideal sites for biofilm formation [4]. Bacteria like Gammaproteobacteria, Alphaproteobacteria, Bacteroidetes, Firmicutes, and Actinobacteria form core communities in biofilms that are well-suited for living with algae [5]. These bacteria, particularly those belonging to the Proteobacteria and Bacteroidetes phyla, are commonly found on macroalgae. These organized bacterial communities create a protective layer on the algal surface, acting as a physical and physiological barrier between the host and its surroundings, thus providing insulation [6]. Although these bacteria are genetically different, they can nevertheless interact and communicate through quorum sensing [7].

Epiphytic bacteria engage in communication and cooperation, both within their species and with other species [8,9]. Additionally, they interact with hosts using chemical signal molecules known as the inter-kingdom QS phenomenon. During QS, bacteria produce and release signals that accumulate until they reach a certain threshold level within their environment [10]. Once this threshold is reached, coordinated activities such as virulence, resistance, bioluminescence, and biofilm formation are regulated [11]. In mutualistic associations between surface bacteria and macroalgae, bacterial secondary metabolites protect the algal host from other organisms and fouling agents in return, benefiting from the readily available organic carbon sources and nutrients produced by the macroalgae [12,13]. However, the relationship between epiphytic bacteria and macroalgae is not always beneficial; some studies have reported potential negative consequences, like the marine macroalga *Delisea pulchra* using a temperature-dependent chemical defense called furanones to protect itself from the bacterial pathogen *Ruegeria* sp. This defense mechanism inhibits bacterial quorum sensing and effectively prevents colonization and infection of *D. pulchra* by R11. Furanones produced by *D. pulchra* hinder the ability of R11 to successfully colonize, as R11 relies on quorum sensing signals for its colonization process [14]. Furthermore, *Acinetobacter* sp. and *Pseudoalteromonas bacteriolytica* are considered causative agents of white rot disease in *Nereocystis luetkeana* and red spot disease in *Laminaria japonica* [15,16,17]. Some other bacterial species, such as *Cobetia marina*, *Pythium porphyrae*, and *Pseudoalteromonas citrea,* have been found to induce diseases in *P. haitanensis* [18,19]. However, the mutually beneficial relationship between adhered bacteria and algae can be broken when bacteria have limited availability of phosphorus during metabolic processes, and both partners may compete for this limited resource [20]. Furthermore, the climatic changes and influx of harmful pollutants impose stress on the marine habitat and its associated microbiota, rendering the bacterial biofilms more susceptible to opportunistic pathogens and consequently induced negative impacts on algae [21].

Biofilms are widely adopted by bacteria in various environments, forming intricate microbial communities through collective behavior and communication systems known as quorum sensing [22,23]. The Quorum sensing involves the production and recognition of signaling molecules called auto-inducers [24,25,26]. Autoinducing peptides (AIPs) and acyl homoserine lactones (AHLs) have been identified as regulators of QS in gram-positive and gram-negative bacteria [27,28,29]. Among various molecular signals utilized in QS systems, the acyl-homoserine lactones (AHLs) or autoinducer type-1 are considered the predominant class [30,31]. AHLs are composed of a homoserine lactone (HSL) moiety linked to fatty acyl chains through an amide bond, and the QS processes relevant to AHL could be common activities in marine environments. Huang and coworkers reported the presence of AI-1 and AI-2 autoinducers in a dinoflagellate bloom and found correlations between dominant species and QS signals, with quorum quenching (QQ) indicating the participation of both QS and QQ in regulating the microbial communities during bloom formation [32]. Moreover, the longer-chain AHLs are believed to be better adapted to marine environments as they seem less affected by pH hydrolysis caused by daily changes in light and dark due to photosynthesis in marine biofilms (Decho et al. 2009) [33]. 

The red macroalga *P. haitanensis*, also known as Zicai, is a widely cultivated edible seaweed in China, particularly in Guangdong, Zhejiang, and Fujian provinces. It holds significant economic importance due to its high nutritional value [34,35]. Most of the studies conducted on *P. haitanensis* focused on its cultivation, secondary metabolite production, and assessment of nutritive values for aquaculture [36,37,38]. However, few studies have reported the impacts of epiphytic bacteria on *P. haitanensis* growth [39]. There exists a study gap because there have been no reports on the presence of AHLs producing epiphytic bacteria associated with the red macroalga *P. haitanensis*. Additionally, the role of AHLs in biofilm formation and their regulatory mechanisms in this context remain unexplored. Therefore, we propose that the epiphytic bacterial community present on the surface of *P. haitanensis* plays a vital role in protecting the host by impeding the settlement and growth of potential competitors. We hypothesize that epiphytic bacterial communities act as an effective additional protective layer by utilizing their QS communication system, thereby providing insulation against host fouling.

To test the hypothesis, epiphytic bacteria from the surface of cultured *P. haitanensis* were isolated to analyze the diverse profile of AHL production and the biofilm-forming capability of these bacteria. Additionally, this study explored the role of AHLs as regulators of biofilm formation. Notably, this is the first study to report the diverse profile of AHL production in *Pseudoalteromonas galatheae* and its role in biofilm regulation. The findings of this study would significantly contribute to our understanding of bacterial colonization on macroalgal surfaces and the involvement of regulators, specifically AHLs. Furthermore, these results serve as a valuable foundation for future investigations into the role of AHLs in the life cycle of *P. haitanensis*.

## 2. Materials and Methods

### 2.1. Sample Collection

*P. haitanensis* was hand-picked using disposable gloves from Nan’ao Island, Guangdong province of China (116°6 40″ E, 23°29′9″ N) in November and March 2018–2019 (Figure 1) during low tides. The collected algal samples were stored in sterile zipper polybags and sealed immediately. Seawater samples were collected from the phycosphere of *P. haitanensis* in sterile plastic bottles. Both the seaweed and seawater samples were transported to the laboratory in cold conditions. During each collection, the temperature, pH, salinity, and dissolved oxygen of the seawater were measured and are presented in Table 1. Upon retrieval, the associated epiphytic bacteria from macroalgal samples were isolated following the established procedure [40]. Briefly, the *P. haitanensis* fronds were gently cleaned with autoclaved seawater at room temperature, and a small portion of the frond was cut using a sterilized blade. The cut portion was then placed onto marine agar 2216 and LB agar and incubated at 25 °C for 2–15 days to isolate the epiphytic bacteria. The plates were observed daily, and different colonies were picked and re-streaked onto the respective media to obtain a pure colony. Each pure bacterial colony was designated with a unique lab code, then maintained at 4 °C in slants and stored in a 25% glycerol suspension in cryotubes at −80 °C for further experiments [40].

### 2.2. Isolation of gDNA and 16S rRNA Gene Amplification

Morphologically different picked bacterial colonies from the previous step were subjected to the isolation of gDNA for 16S rRNA gene amplification. Briefly, gDNA was isolated using a gDNA isolation kit (TIANGEN Biotech, Beijing, China) following the manufacturer’s instructions. The concentration of gDNA was measured using a NanoDrop One (Thermo Fisher, Waltham, MA, USA), and its purification was confirmed with 1% agarose gel electrophoresis. The universal 16S rRNA primers 27F (5′-AGAGTTTGATCMTGGCTCAG-3′) and 1492R (5′-TACGGYTACCTTGTTACGACTT-3′) were used for PCR amplification and sequencing [41]. The reaction mixture and PCR conditions were as follows. The PCR reaction mixture consisted of 2.5 μL of dNTP, 100 ng of each 10× PCR buffer with MgCl2, 25 mM of each forward and reverse primer, 0.5 μL of Taq DNA polymerase, and 10 ng of template DNA. The PCR protocol involved an initial denaturation step at 95 °C for 5 min, followed by 30 cycles at 94 °C for 40 s, 55 °C for 40 s, and 72 °C for 2 min. A final cycle of 10 min at 72 °C was performed using the Bio-Rad T100TM Thermal Cycler (Bio-Rad, Hercules, CA, USA). The amplified products were then analyzed on 1.2% (*w*/*v*) agarose gels stained with ethidium bromide, and the bands were visualized under UV light. For further purification, the PCR products were processed using a QIAquick PCR purification kit (QIAGEN, No. 28104). Subsequently, the amplicons were Sanger sequenced using the 3730XL DNA Analyzer (ABI, Foster, CA, USA).

### 2.3. Bacterial Identification and Phylogenetical Analysis

The sequences were blasted to check their sequence homology against other sequences from NCBI GenBank (https://blast.ncbi.nlm.nih.gov/Blast.cgi) accessed on 15 July 2023. Those with a sequence identity greater than 99% were considered members of the same species [42] (see Table 2). The aligned 16s rRNA bacterial sequences were used to construct the phylogenetic trees with the neighbor joining method [43] using the MEGA 11.0.10 software [44]. The sequences were compiled and aligned using ClustalW embedded in MEGA 11. The Tamura–Nei model was employed to estimate the evolutionary distance model [45]. For reliability, the bootstrap test was performed with 1000 replicates in the phylogenetic trees (Figure 2) [46].

### 2.4. Screening for AHL Production

The bacterial reporter strains used for AHL screening were *Chromobacterium violaceum* CV026, which responds to short carbon chain AHLs (C4–C6), and *Agrobacterium tumefaciens* A136, which responds to long carbon chain AHLs (C6–C12) [47,48].

All the isolated bacteria were screened for AHL production using a 96-well microtiter plate assay [49,50] with slight modifications. In brief, the reporter strain A136 was grown overnight at 28 °C at 150 rpm in 5 mL LB broth supplemented with 50 μg/mL spectinomycin and 4.5 μg/mL tetracycline, while the test bacteria were grown in 75 μL LB broth in a 96-well microtiter plate. The 5× diluted 75 μL of overnight cultured reporter strain A136 was mixed with test bacteria and incubated at 28 °C with constant shaking at 150 rpm for 12 h. X-Gal (40 μg/mL) was added (40 μg/mL) to each well and incubated for 12–24 h. Wells with a blue color were marked as positive. The same protocol was followed with the reporter strain CV026, except for the addition of X-Gal, and wells with purple pigmentation were marked as positive (Figure 3). Positive colonies were double-checked by streaking the bacterial isolate parallel to the reporter strain [51,52]. The pH of 2216E medium was adjusted to 6.7 to avoid the spontaneous alkaline hydrolysis of AHLs during incubation at 28 °C for 24 h.

### 2.5. Isolation and Characterization of AHLs

AHLs were extracted from a positive bacterium colony by individually culturing it in 500 mL of Zobell marine broth pH 6.7 at 28 °C with continuous shaking at 180 rpm for 48 h. Cell suspensions were then aseptically transferred to 50 mL sterile centrifuge tubes and centrifuged for 10 min at 12,000 rpm at 4 °C. The supernatant was collected carefully and mixed with an equal volume of ethyl acetate acidified with acetic acid (0.5%) for liquid-liquid extraction. The process was repeated three times. The upper organic layer was collected separately using a funnel, and the solvent was evaporated using a rotary apparatus to reduce the volume to 1 mL and then completely dried using nitrogen [53,54]. The extracted AHLs were stored at −20 °C for further use.

### 2.6. Identification and Characterization of AHLs by LC–MS

The residues from the previous step were dissolved in 1 mL of HPLC-grade acetonitrile and used for analyzing the samples with liquid chromatography electrospray ionization mass/mass spectrometry (LC–MS). The characteristics of ion products were proposed on the basis of low-resolution MS/MS spectra [55]. The spectra of LC–MS were recorded from 0 *m*/*z* to 300 *m*/*z* to obtain a definite identification of these ion products for their accurate mass values, along with considering the retention time. The theoretical masses of the most likely AHLs in the protonated form were calculated and compared with standards. For our analysis, we used ESI–MS and LC–ESI–MS/MS–CID techniques with a Waters^®^ Micromass^®^ Q-Tof micro™ mass spectrometer connected to a Waters Alliance HPLC and equipped with an electrospray ionization source. During ESI–MS analysis, the samples were directly injected into the mass spectrometry system at a flow rate of 20 μL/min. The capillary voltage was maintained for the sample cone and extraction cone at 2.5 KV, 25 V, and 1.5 V, respectively. For LC–ESI–MS/MS, 2 μL of the sample residues were injected onto a reverse-phase C18 column with a solvent gradient, and argon gas (Phenomenex, 150 mm × 4.6 mm) was used for this process as a collision source.

### 2.7. Biofilm formation Assay

#### 2.7.1. Ninety-Six-Well Microtiter Plate Crystal Violet Biofilm Formation Assay

A biofilm formation assay was conducted in the 96-well microtiter plate as described by [54]. The bacterium was grown in liquid 2216E broth to an OD600 value of 0.5 (approximately 10^7^ CFU/mL in 15–18 h) followed by a 100-fold dilution in the same medium. Subsequently, 200 µL per well of diluted bacteria with exogenous AHLs (i.e., 150 µM of each C8-HSL, 3OC16-HSL, and C18:1-HSL) and without exogenous AHLs were dispensed into the 96-well microtiter plate and incubated at 28 °C for 24 h. *E. coli* DH5 alpha was used as a negative control (NC). The OD 600 of the incubated microtiter plate was measured after every 12 h before washing off the unattached planktonic bacterial cells in order to confirm similar growth of bacteria. Each well was washed three times with 250 µL of 0.1 M PBS (pH 7.3–7.4). The biofilm was fixed with 250 µL of anhydrous methanol by incubating it for 15 min at room temperature, followed by staining with 250 µL per well of 0.1% (*v*/*v*) crystal violet solution. The microtiter plate was incubated at room temperature for 15 min and rinsed three times (250 µL per rinse) with deionized water. The crystal violet was dissolved by the addition of 200 µL of 33% glacial acetic acid, followed by shaking at 150 rpm for 10 min. Finally, the absorbance of the plate at OD 590 was measured using a UV-Vis spectrophotometer.

To measure the effect of different concentrations of selective AHL, i.e., C8-HSL, on biofilm formation based on the previous step, the bacterium was co-cultured with different concentrations, specifically 50 µM, 100 µM, 150 µM, and 200 µM, of exogenous AHLs, and absorbance was measured at OD 590.

#### 2.7.2. Scanning Electron Microscopy Assay

The bacterium P1 (AHL-positive strain) was cultured at 28 °C with shaking at 180 rpm until an OD600 nm value of 1.0 was achieved. Cells were diluted to 1.0 × 10^7^ CFU/mL with culture medium. First, 1 mL of the diluted culture was dispensed onto a pre-treated sterile glass cover slip (0.5 mm × 0.5 mm) placed inside a 24-well microtiter plate and cultured at 28 °C for 24 h. The biofilm deposited on the glass surface was then prepared for scanning electron microscopy (SEM) analysis according to the method described by [56,57]. Plates were wrapped with parafilm in order to avoid evaporation/drying. Later, the broth from the wells was removed gently with a pipette, and a glass coverslip was recovered from the well using clean tweezers, followed by a gentle wash with sterile 0.1 M PBS. The biofilm was fixed with a 2.5% glutaraldehyde solution for 2 h at room temperature, followed by a wash with 0.1 M PBS. The glass coverslip was subjected to dehydration under a graded series of ethanol in ascending order (i.e., 30%, 50%, 70%, 90%, 20 min each) and finally washed twice with 100% ethanol for 20 min. This was followed by critical point drying, platinum sputtering, and SEM observation (JSM 6360L Jeol Tokyo, Japan) performed at 3.0 kV under 5000 and 50,000× magnifications.

#### 2.7.3. Confocal Laser Scanning Microscopy (CLSM) Assay

The overnight bacterial culture was resuspended after 10 min of centrifugation at 7000× *g* in 0.1 M PBS (pH 7.3–7.4) to achieve a final OD600 of 0.25. One ml of the resulting bacterial suspension was loaded into a compartment of a 6-well Petri dish (6 mm Corning Costar 6-well cell culture plate, Greiner Bio-one Corning) with a glass coverslip in the bottom [57]. Notably, to obtain a clean surface of the glass coverslip for biofilm development, glass coverslips were treated with 1 M HCl overnight, followed by washing with anhydrous ethanol and drying at 50 °C for 30 min. The treated coverslips were then placed in the 6-well plate and incubated at 28 °C for 24 h without shaking. The wells were gently washed with 0.1 M PBS (pH 7.3–7.4), and 1 mL of 2216E medium was added to each compartment. Biofilms were then grown for 24 h at 28 °C. The surfaces were then gently rinsed with 0.1 M PBS (pH 7.3–7.4), and biofilms were stained with PI and Syto 9 using the Bact Live/Dead Bacterial Kit (BacLight™ viability kit, Invitrogen, Waltham, MA, USA) in the dark for 15–20 min. After rinsing with ultra-pure water and drying, the samples were analyzed using a Zeiss LSM 800 (Carl Zeiss, Jena, Germany), and 3D structures were reconstituted using software ZEN version 2.3 (Figure 6). COMSTAT 7 built-in with Image J was used for quantification of biofilm formation (Heydron et al. 2000) [58].

Comstat version 2.1 is software designed for analyzing image stacks of biofilms captured using confocal microscopes. It quantifies various factors by breaking down the Z-stack images into small 3D elements called voxels. The voxel’s width (x) and length (y) correspond to the pixel’s sides, while the height (z) represents the spacing between the slices [59]. The following parameters were considered for the quantification of the biofilm formed by *P. galatheae*:Biomass (µm^3^/µm^2^): The volume of biomass per unit area, estimated as the volume of all voxels that contain biomass divided by the substratum area; COMSTAT 2 counts as biomass all voxels above a given threshold.Average thickness (biomass) (µm): Measuring/considering only the area covered by the biomass.Average thickness (entire area) (µm): Measurement of the complete extent of the stack.Maximum thickness (µm): Measurement of the highest point of the biofilm relative to the substratum.Roughness coefficient, Ra (nondimensional): Quantifying the height variability of the biofilm.Surface area (µm^2^): The sum of the areas of all visible biomass voxel surfaces against the background, including the area occupied in each layer (µm^2^), considering biomass pixels in each layer (confocal slices).

### 2.8. Statistical Analysis

A one-way analysis of variance (ANOVA) was conducted to assess the differences in biofilm growth/production by different types of AHLs, considering their respective concentrations. Subsequently, Tukey’s post hoc test was employed to perform pairwise comparisons and identify specific AHL types and concentrations that significantly influenced biofilm growth.

## 3. Results

### 3.1. Identification and Phylogenetic Analysis

Sixteen culturable bacterial strains were isolated from the surface of *P. haitanensis*. The highest number of isolates were selected with 2216E marine agar, followed by LB. Gene amplification (16S rRNA) revealed that the isolates belong to the Vibrio, Pseudoalteromonas, Zobellia, and Firmicutes genera. Vibrio was the highest in number followed by Pseudoalteromonas, Zobellia, and Firmicutes genera. All isolates were found to be associated with marine environments (Table 2).

The phylogenetic tree constructed for the AHL-positive strain represents its closest relatives as obtained from the Gene Bank (Figure 2).

### 3.2. Screening for AHL Production

All isolated bacteria were screened for AHLs using two bio-reporter strains: CV026 for short carbon chain AHLs (C4–C6), and A136 (C6–C14). Initial screening was performed on a 96-well microtiter plate. Out of sixteen isolates, only one isolate (i.e., P1), which was later identified as *Pseudoalteromonas galatheae*, produced indigo (blue color) among all the test strains when incubated at 28 °C for 24 h, demonstrating the production of AHLs (Figure 3). This positive bacterial isolate was confirmed for AHL production by parallel streaking on 2216E marine agar to make double confirmation.

### 3.3. Extraction and Characterization of AHL

LC–MS data of the strain P1 extract show the presence of a characteristic lactone fragment at *m*/*z* of 102 and the molecular ion peak at *m*/*z* of respective ions, suggesting different AHL production. The results revealed that our isolate could produce four different types of QS signals, i.e., C4-HSL, which had a precursor ion (*m*/*z*) of 172.2 and a retention time of 5.80 min. For C8-HSL, the precursor ion (*m*/*z*) was 228.2 and retention time was 4.72 min, and for 3-oxo-C16-HSL and C16-HSL, the precursor ion (*m*/*z*) was 214.1 and 368.3 and retention time was 8.0 and 9.03 min, respectively (Figure 4).

### 3.4. Biofilm Formation Assay

#### 3.4.1. Crystal Violet Assay

Crystal violet assay revealed that the growth of *P. galatheae* was not affected by exogenous AHLs, i.e., C4-HSL, C8-HSL, 3OC16, and C18:1-HSL. The growth curves show almost similar patterns with and without exogenous AHLs in the growth media, suggesting no role for AHLs in the growth of bacterial cells (Appendix A). Additionally, no correlation was found between growth and different AHLs (Appendix A).

However, a significantly higher level of biofilm formation was observed in the presence of C8-HSL. Other exogenous AHLs, such as C4-HSL, 3OC16, and C18:1, also enhanced the formation of biofilm compared to biofilm formation by *P. galatheae* without exogenous AHL (Figure 5A,C). This suggests that C8-HSL primarily regulates biofilm formation in *P. galatheae*. Moreover, it was also observed that higher biofilm was produced with increasing concentration (50 µM–200 µM), demonstrating a positive correlation between C8-HSL concentration and biofilm formation (Figure 5B,D and Appendix A).

#### 3.4.2. Scanning Electron Microscopy (SEM) Assay

Scanning electron microscopy of the biofilm on a glass cover slip revealed that *P. galatheae* formed biofilm with and without exogenous HSL. In the absence of C8-HSL, a thin biofilm layer was scattered all over the cover slips during the initial, intermediate, and final stages (Figure 6a–c). However, cells treated with 200 µM exogenous C8-HSL formed a highly dense biofilm that covered most of the glass coverslip during the initial, intermediate, and final stages (Figure 6d–f). Moreover, a thick EPS matrix was also observed with bacteria trapped in it. This observation validates the results of the crystal violet assay regarding biofilm formation and its regulation by AHLs.

#### 3.4.3. Confocal Laser Scanning Microscopy (CLSM) Assay

The biofilm developed on a standard glass coverslip by *P. galatheae* in the presence and absence of exogenous AHLs was studied using Confocal Laser Scanning Microscopy (CLSM). The images from CLSM and biofilm quantification are presented in Figure 7 and Table 3, respectively. The results revealed high biofilm formation in the presence of exogenous AHLs (200 µM of C8-HSL). The effects of AHLs on biofilm characteristics were investigated using Comstat version 2.1 integrated with ImageJ. The analysis revealed significant differences between the “with AHL” and “without AHL” conditions. In the absence of AHL, the biomass of both dead and live cells was lower, with corresponding mean thickness values. However, the presence of AHL significantly increased biomass, mean thickness (both biomass and area), and maximum thickness. Moreover, AHL was found to decrease surface roughness, resulting in a smoother biofilm surface. Additionally, AHL positively influenced the expansion of the biofilm surface area. These findings highlight the role of AHL in promoting biofilm growth, thickness, and surface properties, suggesting its importance in shaping the structural characteristics of biofilms (Figure 7, Table 3).

## 4. Discussion

Marine macroalgae establish highly specific associations with numerous microorganisms. Bacteria inhabiting the macroalgal surface have been found to utilize AHL-QS systems for their cell-to-cell communication, which often plays a crucial role in bacterial colonization, such as biofilm formation. This symbiotic relationship between bacteria and their hosts was found to be regulated by AHL quorum sensing [60,61,62]. The bacterial communities identified in this study were similar to those identified in different seaweeds [63,64]. Gammaproteobacterial genera, including Vibrio, Pseudomonas, and Acinetobacter, are known for their well-studied AHL-based QS systems. These bacteria are commonly found in different marine habitats and are known for producing AHL molecules [65,66,67,68,69,70,71,72]. Additionally, metagenomics studies reported that Vibrionales, Pseudomonadales, or Alteromonadales are the main contributors to AHL production [32,63,64,65,66,67,68,69,70,71,72,73,74,75]. Furthermore, it has been demonstrated that Gammaproteobacteria AHL pathways regulate various traits that may have significance in biogeochemical processes, including biofilm regulation, production of hydrolytic enzymes, siderophores, and motility [48,49].

Most of the Pseudoalteromonas species obtained from living surfaces are often found in epiphytic and epizoic microbiomes associated with marine microorganisms [76]. Pseudoalteromonas isolates are frequently recognized as valuable sources of bioactive exoproducts, specifically secondary metabolites like exopolymeric substances and extracellular enzymes [77]. However, there is a paucity of knowledge on cultured *P. haitanensis* macroalga-associated microbial communities with the QS phenotype. Therefore, this study was aimed at surveying, for the first time, the QS activity from the epiphytes of *P. haitanensis*, along with biofilm-forming capability and its regulation.

We isolated *Pseudoalteromonas galatheae* from the surface of *P. haitanensis* (Figure 2) with the ability to produce QS signals and exhibit a biofilm-forming phenotype (Figure 4). For the AHL screening assay, CV 026 and A136 reporter strains were used, with the biosensor CV026 containing a deletion in the luxI homolog cviI gene, which produces the purple pigment violacein in response to fully reduced short-chain AHLs (C4–C6 side chains) with a hydrogen as the R-group at the b carbon [78]. The A136 report strain expresses a lacZ fusion most strongly in response to medium-chain-length AHLs (C6–C12, although weakly to C4), with limited distinction of AHLs carrying a hydrogen, a hydroxyl, or a carbonyl as the R-group at the b carbon as reflected by β-galactosidase activity [47,79,80] (None of the epiphytic bacteria respond to CV026. However, the strain (PI) identified as *P. galatheae* demonstrates β-galactosidase activity (Figure 3).

Further characterization of the bacterial culture extract by LC–MS data revealed that *P. galatheae* produces four types of AHL signal molecules, including three linear chain AHLs (C4, C8, and C18-1), and one β-carbon-oxidative AHL product i(3OC6), with the following concentration order: C4 > C8 > 3OC6 > C18:1 (Figure 4). Our findings are consistent with previous research demonstrating the production of diverse AHLs by individual bacterial species. For instance, *Pseudoalteromonas* sp. R3, obtained from marine sediment, could produce two AHLs, namely 3-OH-C6-HSL and C8-HSL [81]. Similarly, *Pantoea ananatis* B9, isolated from natural marine snow particles in marginal seas of China, exhibited the ability to produce six different AHLs: C4-HSL, 3OC6-HSL, C6-HSL, C10-HSL, C12-HSL, and C14-HSL [61]. Marine bacterial strains ST2 from *Scrippsiella trochoidea* and RT1 from the root nodules of *Lens culinaris*, have been found to produce multiple AHL compounds, including C4-AHL, C8-AHL, and C10-AHL, and 3-oxo-C8-HSL and 3-OH-C10-HSL, respectively [75]. Furthermore, 3-oxo-C8-HSL, produced by RT1, has also been found to influence bacterial motility and biofilm formation [82]. Additionally, the bacterium *Anemonia viridis*, isolated from cnidarian species, has been reported to produce five AHLs: C6-HSL, C8-HSL, 3-OH-C6-HSL, 3-OH-C8-HSL, and 3-OH-C10-HSL [83]. Certain Pseudoalteromonas strains, such as 520P1 and NJ6-3-1, have also been shown to produce AHLs like 3-oxo-C8HSL, C14-HSL, and C8-HSL, respectively, and have regulatory effects on violacein production and the secretion of antibacterial compounds [84,85]. In another study, Ziesche and colleagues reported that marine Roseobacter clade bacteria isolated from macroalgae could produce a variety of AHLs [86]. Moreover, *P. ananatis* SK-1, isolated from the Shirakawa River in Japan, could produce two AHLs, namely C6-HSL and 3OC6-HSL, and induce center rot disease in onions using an AHL-based QS system [87]. Another bacterium, *Pseudomonas fluorescens* PF07, known for fish spoilage, produces three different AHLs, with C4-HSL being the predominant signal molecule and acting as a biofilm regulator [88]. It has been revealed that the production of various long-chain AHLs is species-specific, and the bacterial community associated with *A. viridis* undergoes compositional changes alongside AHL profiles, indicating a potential connection between bacterial community dynamics and quorum sensing [83]. Epiphytic bacterium *Shewanella algae* of red macroalgae have been found to produce five types of AHLs, namely C4-HSL, HC4-HSL, C6-HSL, 3OC6-HSL, and 3OC12-HSL, and induce carpospore liberation in *Gracilaria dura* [89]. It is suggested that bacteria produce long-chain AHLs as an adaptive response to the alkaline seawater environment [52]. Therefore, we assume that the epiphytic bacterium *P. galatheae* produces a variety of AHLs, mostly long chains, as an adaptive response to the alkaline seawater environment.

Bacteria utilize QS as part of their genetic machinery, enabling them to dominate marine niches and drive the succession of the entire bacterial community [52]. Bacterial biofilm formation and its regulation have been reported to be controlled by QS [88,90]. The biofilm acts as a defensive coating, protecting the surface of macroalgae from macrofoulers [91]. We also observed a biofilm-forming phenotype in our isolated bacterium, *P. galatheae* (Figure 5). Our data on biofilm formation by *P. galatheae* and its regulation by AHLs demonstrate similar results to those reported previously [47,92,93]. It is well established that the gene encoding proteins involved in biofilm formation are regulated by the QS system [94]. Several bacterial species, such as *Pseudomonas putida*, *Pseudomonas fluorescens*, *Pseudomonas aeruginosa*, and *Burkholderia cenocepacia*, have been studied for their regulation of biofilm formation, with observations suggesting that biofilm formation is governed by three regulatory systems, i.e., c-di-GMP, sRNA, and QS [95]. In contrast to Tang’s study [88], our data revealed that C8-HSL played a significant role as the primary biofilm regulator (Figure 5) instead of C4-HSL. Whereas the study of Hayek and co workers reported that C8-HSL had no impact on biofilm adhesion but regulates bioluminescence in the marine bacterium *Shewanella woodyi* [96]. However, our results are similar to those of previous study, where diatom biofilm thickness was enhanced in the presence of C8-HSL along with other AHLs [97]. Additionally, C8-HSL significantly increased the biomass and thickness (both by biomass and surface area) of the biofilm formed [97]. Similarly, *P. galatheae* also demonstrates similar biofilm structure formation as revealed by the biofilm CLSM assay and biofilm quantification (Figure 7, Table 3). Interestingly, while conducting co-culture experiments with different AHLs, bacterial growth demonstrates a similar growth curve (Appendix A), yet the production of biofilm was higher in the presence of exogenous AHLs, as confirmed by crystal violet, SEM, and CLSM assays (Figure 4, Figure 5, and Figure 6), further supporting the argument that C8-HSL is the primary regulator of biofilm formation in *Pseudoalteromonas galatheae*. The positive correlation between different concentrations of C8-HSL and biofilm formation revealed that relative increases in the concentration of up to 200 μM also enhance biofilm formation (Appendix A).

However, to gain further insight into the mechanism of AHLs and their role in biofilm formation and regulation in macroalga, more studies, especially concerning biofilm gene expression, are warranted. However, the specific physiological role of AHL molecules within this epiphytic bacterium is yet to be fully understood. To gain a comprehensive understanding of AHL producers and their ecological functions within the macroalgal ecosystem, further investigation using reporter strains and non-cultivable approaches targeting AHL genes is warranted. This study contributes valuable insights into the cultivable population of AHL-producing bacteria associated with *P. haitanensis*. The findings from this study will significantly advance future research on the roles of AHL signaling molecules in macroalgal-epiphytic interactions and the role of AHLs in the life cycle of host macroalga *P. haitanensis.*

## 5. Conclusions

The cultural diversity of epiphytic bacteria in the cultured red macroalga *P. haitanensis* revealed a predominance of Vibrio species, followed by Pseudoalteromonas. Among these bacteria, a specific strain named *Pseudoalteromonas galatheae*, isolated from *P. haitanensis*, produces four types of AHL signal molecules, namely C4-HSL, C8-HSL, C18-HSL, and 3-oxo-C16-HSL. Additionally, this bacterium also exhibited a biofilm-forming phenotype. The regulation of biofilm formation was observed to be influenced by quorum sensing signal molecules, particularly C8-HSL, and show a positive correlation with its concentration. This is the first report of QS activity from epiphytic bacteria of *P. haitanensis* in an artificial culture environment. These findings provide valuable insights for designing future investigations that explore the role of AHL-producing epiphytes in the life cycle of *P. haitanensis.*

## Figures and Tables

**Figure 1 microorganisms-11-02228-f001:**
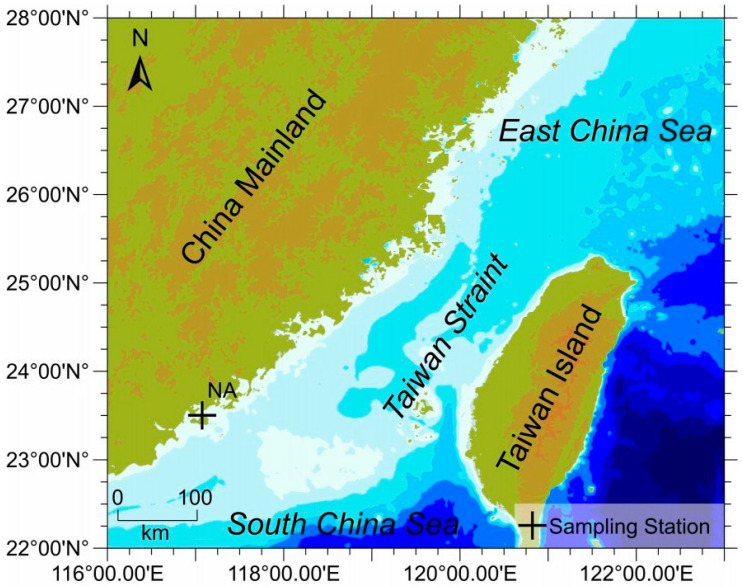
Map of the study area with sampling sites for macroalgae *P. haitanensis* cultures. The + indicates the location of the *P. haitanensis* culture site, from where samplings were collected.

**Figure 2 microorganisms-11-02228-f002:**
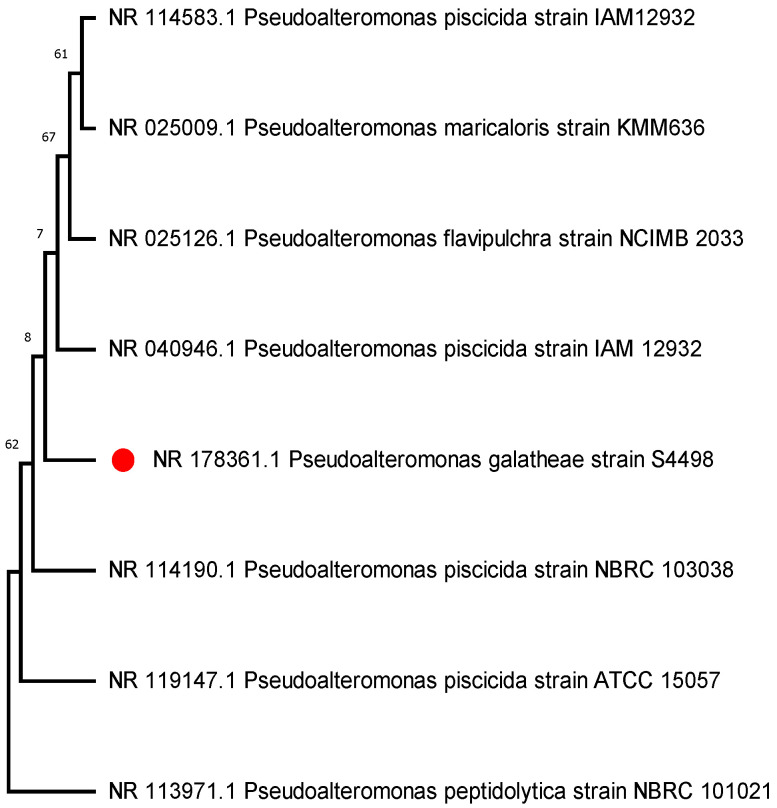
Phylogenetic tree of the AHL positive strain (red dot) created with MEGA11 using the Neighbor–Joining method. Eight nucleotide sequences were analyzed, and bootstrap values (based on 1000 replicates) indicating the support for each branch are shown. Evolutionary distances were computed using the Maximum Composite Likelihood method, and all ambiguous positions were removed for each sequence pair (pairwise deletion option).

**Figure 3 microorganisms-11-02228-f003:**
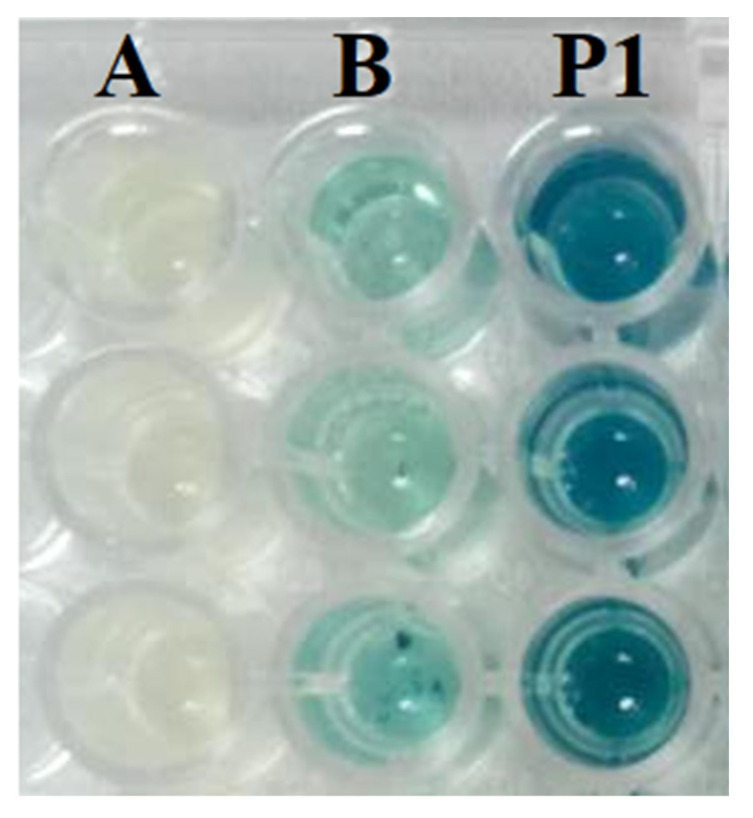
(A) Negative control A136 + X-Gal, (B) positive control A136 + X-Gal + AHL, and (P1) test strain + A136 + X-Gal showing indigo (blue) coloration. Data were obtained in triplicate.

**Figure 4 microorganisms-11-02228-f004:**
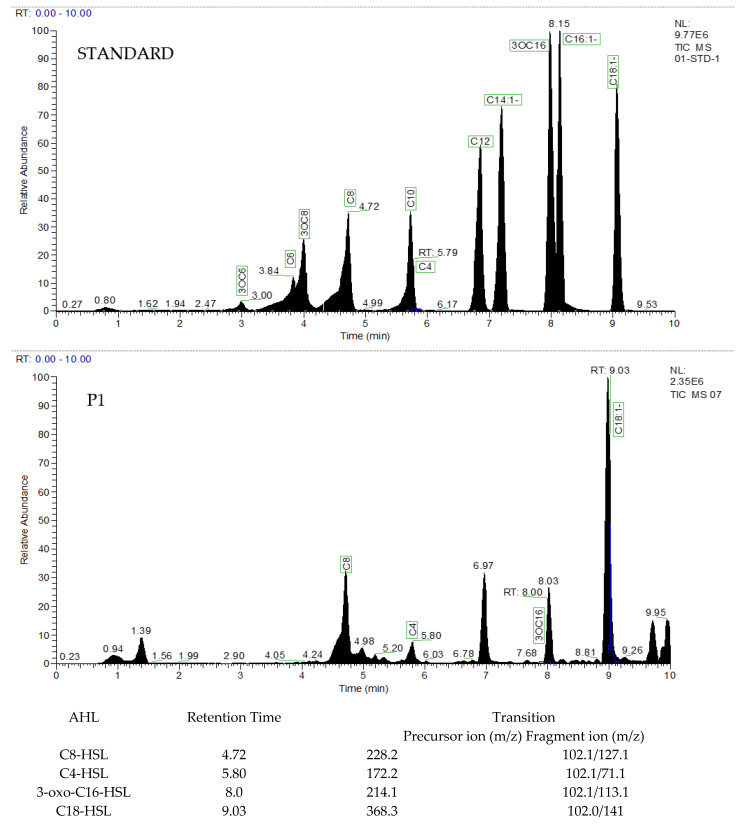
LC–MS graph showing peaks of different AHLs detected compared to the standard. AHLs from PI and their respective retention times, precursor ion (*m*/*z*) values, and fragment ion (*m*/*z*) values are presented in the table.

**Figure 5 microorganisms-11-02228-f005:**
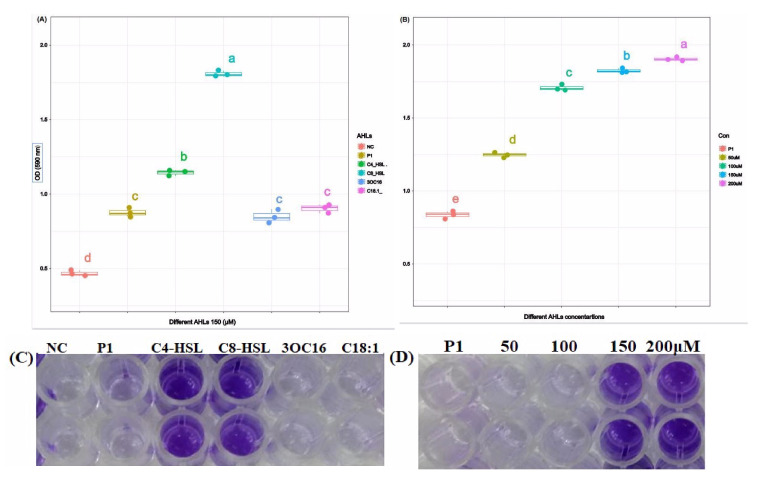
(**A**,**C**) *P. galatheae* bacterium growth in the absence and presence of different exogenous AHLs. (**B**,**D**) Biofilm formation in the presence of different concentrations of exogenous C8-HSL. Distinct letters on boxplots indicate significant differences among various concentrations (a) and doses (b), as revealed by Tukey’s post hoc test (*p*-value = 0.5) using R.

**Figure 6 microorganisms-11-02228-f006:**
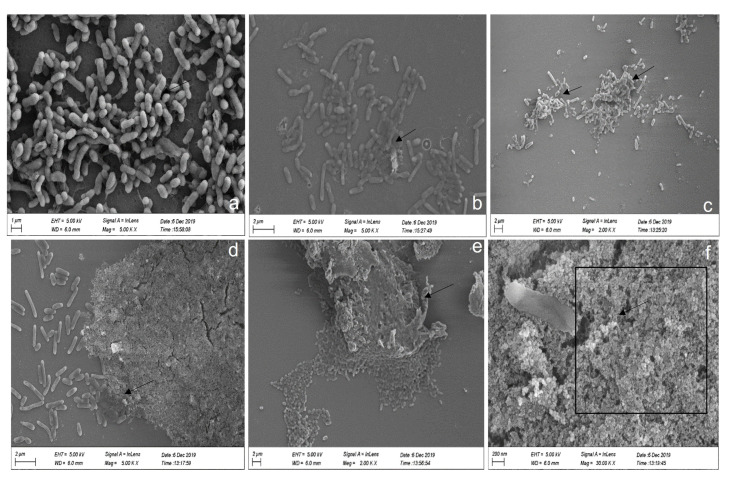
SEM of Biofilm formation by *P. galatheae* on a glass cover slip. (**a**) Initial stage of biofilm formation without C8-HSL. (**b**) Intermediate stage of biofilm formation without C8-HSL. (**c**) Final stage of biofilm formation without C8-HSL. (**d**) Initial stage of biofilm formation with C8-HSL. (**e**) Intermediate stage of biofilm formation with C8-HSL. (**f**) Final stage of biofilm formation with C8-HSL. Without C8-HSL, a thin biofilm is observed, while with C8-HSL, a much thicker and denser biofilm is formed. Arrows represent the biofilm formation and square represents thick biofilm.

**Figure 7 microorganisms-11-02228-f007:**
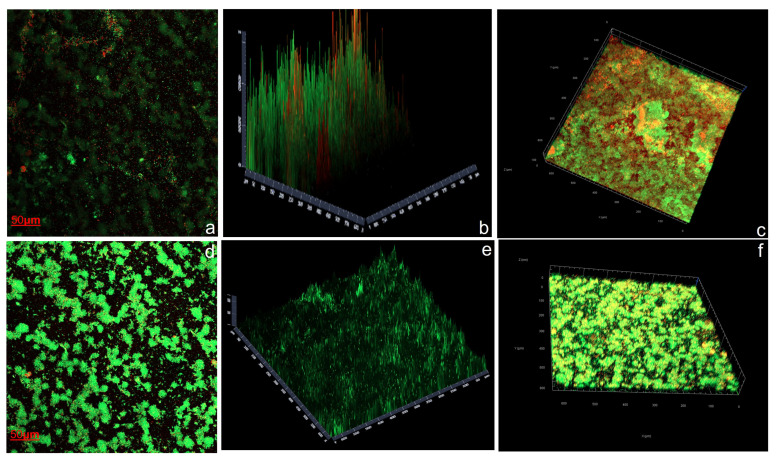
CLSM of biofilm formation by *P. galatheae* on a glass cover slip. (**a**) A 2D structure of biofilm without C8-HSL, with dead cells represented in red and live cells in green. (**b**) A 2.5D structure of biofilm without C8-HSL. (**c**) A 3D image of biofilm without C8-HSL. (**d**) A 2D structure of biofilm in the presence of C8-HSL. (**e**) A 2.5D structure of biofilm in the presence of C8-HSL. (**f**) A 3D structure of biofilm in the presence of C8-HSL. The biofilm in the absence of C8-HSL shows an uneven and thin surface with a higher number of dead cells. However, in the presence of C8-HSL, the biofilm appears smooth, healthy, and dense. Green color represents live and red color represents dead bacterial cells during biofilm formation.

**Table 1 microorganisms-11-02228-t001:** Physical parameters of the sampling site, including the mean of three replicates.

Location (Nan’ao)				
Month	Temperature (°C)	pH	Salinity (ppt)	DO
November	13.8	7.8	30.5	8.0
March	21.2	8.1	31.7	8.1

**Table 2 microorganisms-11-02228-t002:** Isolated culturable epiphytic bacteria and their similarity with previously reported marine bacteria.

Lab Code	Identified as	Accession No.	Similarity (%)	Genera
P1	*Pseudoalteromonas galatheae*	NR_178361.1	99.59	Proteobacteria
P2	*Staphylococcus saprophyticus* strain UTI-045	CP054831.1	100.00	Firmicutes
P3	*Vibrio alginolyticus* strain NBRC 15630	NR_122050.1	99.55	Proteobacteria
P4	Vibrio sp. strain 201709CJKOP-32	MH093789.1	99.48	Proteobacteria
P5	*Vibrio antiquarius*	NC_013456.1	98.82	Proteobacteria
P6	Vibrio sp. strain 201709CJKOP-38	MH093795.1	99.93	Proteobacteria
P8	*Zobellia russellii* strain KMM 3677	NR_024828.1	100	Bacteroidetes
P9	*Paraglaciecola mesophila* KMM 241	NZ_BAEP01000046.1	99.70	Proteobacteria
P10	*Vibrio diabolicus* strain HE800	NR_036811.1	98.82	Proteobacteria
P12	Vibrio sp. VibC-Oc-059	KF577068.1	100.00	Proteobacteria
P15	Pseudoalteromonas sp. strain 5315	ON026012.1	100	Proteobacteria
P17	*Pseudoalteromonas distincta* strain KMM 3548	NR_025654.1	99.7	Proteobacteria
P18	Pseudoalteromonas sp. ZJHD1-34	JN107745.1	99.77	Proteobacteria
P19	Vibrio sp. strain *201709CJKOP-60*	MH093817.1	99.92	Proteobacteria
P20	*Vibrio alginolyticus* strain 2014V-1011	CP046772.1	100.00	Proteobacteria
P21	*Halomonas venusta* strain MA-ZP17-13	CP034367.1	99.80	Proteobacteria

**Table 3 microorganisms-11-02228-t003:** Biomass, mean thickness, roughness, and surface area of biofilm growth on glass cover slip in the presence and absence of C8-HSL. Analysis was performed using COMSTAT. The values are the mean ± standard deviation of the data from five replicates.

*P. galatheae*	Without AHL		With AHL	
	Dead	Live	Dead	Live
Biomass (µm^3^/µm^2^)	1.14 ± 0.01	2.63 ± 0.37	5.91 ± 0.78	9.99 ± 1.30
Mean thickness (Biomass) (µm)	4.36 ± 0.60	6.50 ± 1.25	7.42 ± 1.20	9.60 ± 0.94
Mean thickness (Area) (µm)	1.24 ± 0.03	4.36 ± 1.05	5.05 ± 1.31	8.10 ± 1.37
Maximum thickness (µm)	5.54 ± 0.52	8.61 ± 1.88	21.01 ± 4.18	24.05 ± 6.24
Roughness (–)	1.85 ± 0.03	0.95 ± 0.22	0.76 ± 0.29	0.19 ± 0.18
Surface Area (10^5^ m^2^)	0.22 ± 0.02	1.81 ± 0.44	2.90 ± 0.71	3.76 ± 0.33

## Data Availability

Not applicable.

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
