# Peer review of "Unraveling the Diverse Profile of N-Acyl Homoserine Lactone Signals and Their Role in the Regulation of Biofilm Formation in Porphyra haitanensis-Associated Pseudoalteromonas galatheae"

_microorganisms, 2023, doi:10.3390/microorganisms11092228_

Round 1

Reviewer 1 Report

The authors found a specific strain named Pseudoalteromonas galatheae, isolated from P. haitanensis, produces four types of AHLs signal molecules, namely C4-HSL, C8-HSL, C18-HSL, and 3-oxo-C16-HSL. Additionally, this bacterium also exhibited a biofilm-forming phenotype. and it was observed that its biofilm formation phenotype is regulated by QS molecules, particularly C8-HSL. 

Some minor points:

Line 221. 2.7.2. & line 320. 3.4.2. subheading. Typo. Should be "Scanning Electron Microscopy"

Line 290. 3.3. Extraction and Characterization of AHL. Not clear how many replicates have been done? Are they consistent? 

Table 3. First column the measurement items are confused. Please check.

Author Response

Dear Reviewer

We are grateful for the thorough evaluation of our manuscript and the insightful comments provided by you. We have carefully considered each of your suggestions, queries, and recommendations, and we tried our best to address them in a comprehensive manner.

In response to your invaluable feedback, we have provided detailed point-to-point answers to each of your queries in the attached PDF. We believe that these responses effectively address the concerns raised during the review process. Additionally, we have made the necessary revisions to the manuscript based on your feedback to enhance the clarity, accuracy, and overall quality of the content.

We are confident that the revised manuscript, along with the accompanying responses, adequately addresses the issues highlighted by you.

We genuinely appreciate the time and effort you have invested in evaluating our work, and we would like to express our gratitude for your continued support in advancing the knowledge in our field. We are committed to ensuring that the manuscript meets the standards of excellence set by the journal.

We look forward to your assessment of our revised submission. Please do not hesitate to reach out if you have any further questions or require additional information.

Sincerely,

Prof. Du Hong

Reviewer 2 Report

The work presented by Aslam et al. provides interesting and innovative information about bacterial population found in macroalgae Porphyra haitanensis. The introduction gives relevance of this algae and the importance of biofilm forming bacteria attached to it. Also, quorum sensing analysis gives an insight about the molecular processes affecting the proliferation of both bacterial population and algae. Some important points that must be attended are mentioned below.

Introduction: At the end of paragraph one, authors mention the benefits of biofilm formed by epiphytic bacteria, but later, in line 54, it is mention that this relationship is always not beneficial, and an example of the pathogen Ruegeria sp. R11 is used. It is a little confusing, probable due to the term “always not beneficial”. It is suggested to check if the idea that authors wanted to express is correct.

Line 129: Table 1, complete the line under DO.

Line 139: Correct the formula writing, also, include a space between 25 and mM.

Line 199: Missing final point.

Line 201: Eliminate colon

Line 203: OD600 implies that measurement was at 600 nm, it’s not necessary to include the nm.

Methodology is complete and understandable, but writing must be improved. A lot of errors in units (number and unit without spacing), points and comas wrongly used were detected. Please check again and correct.

Results, several bacterial strain names are not in italic.

Figure resolution must improve. For example, figure 3 and 5, although visible, are completely pixeled.

Use of “i.e.” should be limited.

Line 339: It is mentioned that results are shown in table 2, but this doesn’t correspond to the information.

In discussion, bacteria names are not in italic thought the section. Please check and correct.

Line 428. Missing a capital letter after the 84 reference.

Check spelling and writing. Some parts have writing and conjugation errors.

Author Response

Dear Reviewer

We are grateful for the thorough evaluation of our manuscript and the insightful comments provided by you. We have carefully considered each of your suggestions, queries, and recommendations, and we tried our best to address them in a comprehensive manner.

In response to your invaluable feedback, we have provided detailed point-to-point answers to each of your queries in the attached PDF file titled "Response to Reviewer's Comments." We believe that these responses effectively address the concerns raised during the review process. Additionally, we have made the necessary revisions to the manuscript based on your feedback to enhance the clarity, accuracy, and overall quality of the content.

We are confident that the revised manuscript, along with the accompanying responses, adequately addresses the issues highlighted by you.

We genuinely appreciate the time and effort you have invested in evaluating our work, and we would like to express our gratitude for your continued support in advancing the knowledge in our field. We are committed to ensuring that the manuscript meets the standards of excellence set by the journal.

We look forward to your assessment of our revised submission. Please do not hesitate to reach out if you have any further questions or require additional information.

Sincerely,

Prof. Du Hong

Reviewer 3 Report

Title: Unraveling the Diverse Profile of N-Acyl Homoserine Lactones Signals and their Role in the Regulation of Biofilm Formation in Porphyra haitanensis associated Pseudoalteromonas galatheae 

As stated in the abstracts “N-acyl homoserine lactones (AHLs) are small diffusible chemical signal molecules that serve as social interaction tools for bacteria, enabling them to synchronize their collective actions in a density-dependent manner through quorum sensing (QS)”.

This study is about QS activity from epiphytic bacteria of red macroalgae Porphyra haitanensis, along with its involvement in biofilm formation and regulation” [by the epiphytic species Pseudoalteromonas galatheae] and authors are looking at the role of AHLs in biofilm formation and their regulatory mechanisms).

The epiphytic bacterial community acts as an effective additional protective layer by utilizing their QS communication system, thereby providing insulation against host fouling.

Methods: Uses AHL reporter strains, Agrobacterium tumefaciens A136 and Chromobacterium violaceum CV026. Authors use crystal violet assay to look for “biofilm forming” phenotype. They should really be looking for “thin biofilm” phenotype that still forms biofilms, but this type sheds off progeny by exfoliation and avoids making the biofilm thicker and thicker. The advantage is the faster growth rate from advective transport and supply of nutrients which won’t penetrate into the inner layers of the biofilm. This sends them into what you call stationary or decline phase. You need an additional model that measures the growth rate which can be controlled by continuous flow of nutrient medium. Species that can colonize surfaces and grow as a monolayer have more interesting properties than the phenotypes that cause biofilms to continue to get thicker and denser and starve their inner layers of cells deep in the middle.

2.1. Sample Collection; P. haitanensis was hand-picked. Figure 1 shows map of the study area with sampling site of macroalgae P. haitanensis culture and Table 1 shows physical parameters of the sampling site. Mean of three replicates (good).

2.2. Isolation of gDNA and 16S rRNA Gene Amplification:

2.3. Bacterial Identification and Phylogenetical Analysis and 2.4 Screening for AHL Production and 2.5 Identification and Characterization of AHLs by LC-MS: These sections are fine.

2.6.Fine

2.7. Biofilm formation Assay 200

2.7.1. 96-Well Crystal Violet Biofilm Formation Assay:

2.7.2. Scan Electron Microscopy Assay (on glass cover slips)

2.7.3. Confocal Laser Scanning Microscopy (CLSM) Assay

You do not have a proper kinetic flow model for testing phenotypes and genotypes? Why don’t you use a growth rate-controlled model (e.g. perfusion matrix model)?

2.8. Statistical Analysis: These seem appropriate.

Results:

3.1. Identification and Phylogenetic analysis, Table 2 shows isolated culturable epiphytic bacteria and their similarity with previously reported marine bacteria.

3.2. Screening for AH (using bioreporter species) (Figure 3). These sections are very sound

3.3. Extraction and Characterization of AHL: LC-MS data: very good data.

Figure 4 shows LC-MS graph showing peaks of different AHLs detected compared to standard.

3.4. Biofilm Formation Assay. Firstly 3.4.1. Crystal Violet Assay. You state

3.4.2. Scan Electron Microscopy (SEM) Assay

Figure 6. SEM of Biofilm formation by P. galatheae on glass cover slip. Without C8-HSL, a thin biofilm is observed: This is an interesting finding: Biofilms that remain thin also remain well fed since they have no problem with diffusion limitation. Monolayer biofilms grow and shed progeny. Your reporting of growth of the biofilm does (or does not) count the cells that are being shed from the biofilm into the liquid planktonic state and properly count the populations and express the numbers of cells per square cm. What is the growth rate of your growing biofilms. This I do not see.

Table 3. and figure 7 shows CLSM of biofilm formation on glass cover slip (a) 2DBiomass, mean thickness, roughness, and surface area of biofilm growth on glass cover slip in the presence and absence of C8-HSL. Analysis was performed using COMSTAT. Values mean ± standard deviation of data from 5 replicates.

The inferences revealed that P. galatheae produces four different types of AHL molecules such as i.e., C4-HSL, C8-HSL, C18-HSL and 3-oxo-C16-HSL and it was observed that its biofilm formation phenotype is regulated by QS molecules. This is the first study providing insights into the QS activity, diverse AHL profile, and regulatory mechanisms that govern the biofilm formation phenotype of P. galatheae. These findings offer valuable insights for future investigations exploring the role of AHL producing epiphytes and biofilms in the life cycle of P. haitanensis.

This suggests no role of AHLs in the growth of bacterial cells (Fig S1). In addition no correlation was found between growth and different AHLs (Fig S2). Have you been accurately monitoring the growth rate in any accurate way? You have not shown any growth curves nor made reference to growth rate.

Without C8-HSL, a thin biofilm is observed, while with C8-HSL, a much thicker and denser biofilm is formed. Is the thin biofilm due to (1) deceleration of the growth rate of the attached layer (i.e. more stationary phase cells) or is the attached monolayer (or thin layer) producing progeny which become unattached (i.e. planktonic and flows away; they desquamate compared to the monolayer (or mother layer). Such thin biofilms are reported to grow faster because the inner cells always get a good supply of nutrients because the biofilm is too thin to cause diffusion limitation. In thick biofilms new progeny stay attached. When they reach a thickness of about >20-30 micrometers, then diffusion limitation of the growth-limiting element starts to occur. This is particularly the case if carbon-energy sources are at high concentrations and extracellular production of polymers or gels then increase the biofilm density as well as its area and depth. The biofilm grows thicker and there is considerable diffusion resistance (i.e. lack of penetration of nutrients into the inner layers) which slows metabolism and growth rate of the biofilm. The technique of perfusion matrix biofilm model (a method not employed by the present authors) grow target microbes as attached cells on a loose matrix with a relatively fast flow of feedstock. Such systems show biofilm steady states at growth rates controlled by the flow rate. The model assumes moderate to high shear rates to help ensure that new progeny produced by the biofilm are desquamated off the surface of the mother layer so that the mother layer population remains constant and its growth rate is measured by the rate that desquamated cells leave the reactor. See “flat surface biofilm model by Slade EA, Thorn RMS, et al., (2019). An in vitro collagen perfusion wound biofilm model; with applications for antimicrobial studies and microbial metabolomics. BMC Microbiol. 19; (1):310.

This perfusion model has the advantage of the researcher fixing the growth rate by controlling the flow rate and supply rate of growth medium coming into the system. Growth rate-controlled conditions are the only way of seriously studying this subject.

Other questions: What is the role of cyclic di-GMP? And role of Diguanylate cyclases/ phosphodiesterases (if any) in quorum sensing? Or have they no connections? The first messengers are extracellular signalling molecules such as nutrient substrate. The second messengers are intracellular signalling molecules released by the cell in response to the first messenger. I thought quorum sensing involved the global secondary messenger, c-di-GMP? Yet this is only mentioned once in the whole script.

The cyclic dinucleotide second messenger: c-di-GMP was first shown in 1987 to regulate cellulose synthesis in Acetobacter xylinum. Now demonstrated to be involved in EPS production and is a key player in the transition from motile planktonic to sedentary biofilm modes. Reduction of c-di-GMP has been shown to reduce Pseudomonas biofilms whilst increased c-di-GMP, increases biofilm formation.

Do different QS manifest via secondary messengers? Both quorum sensing and c-di-GMP are strongly associated with the bacterial cell growth rate. Without growth rate controls the interpretation of biofilm behaviour is uncertain. Perhaps the authors should reflect that growth rate “by itself” (i.e. independently) affects microbial physiology and causes significant gene control (repression, de-repression, induction and lots of phenotypic switching). This reviewer thinks that research into regulation without controlling for growth rate is a significant weakness.

Otherwise, the research is good to excellent in places.

The manuscript can certainly be improved and more explanation is required to understand how QS fits in, into existing secondary messaging using c-di-GMP

Author Response

Dear Reviewer

We are grateful for the thorough evaluation of our manuscript and the insightful comments provided by you. We have carefully considered each of your suggestions, queries, and recommendations, and we tried our best to address them in a comprehensive manner.

In response to your invaluable feedback, we have provided detailed point-to-point answers to each of your queries in the attached PDF file. We believe that these responses effectively address the concerns raised during the review process. Additionally, we have made the necessary revisions to the manuscript based on your feedback to enhance the clarity, accuracy, and overall quality of the content.

We are confident that the revised manuscript, along with the accompanying responses, adequately addresses the issues highlighted by you.

We genuinely appreciate the time and effort you have invested in evaluating our work, and we would like to express our gratitude for your continued support in advancing the knowledge in our field. We are committed to ensuring that the manuscript meets the standards of excellence set by the journal.

We look forward to your assessment of our revised submission. Please do not hesitate to reach out if you have any further questions or require additional information.

Sincerely,

Prof. Du Hong

Round 2

Reviewer 3 Report

This manuscript has been much improved, and many of the queries that I raised have been answered.

I have no further comments.